# Quality of Life in Romanian Children with Type 1 Diabetes: A Cross-Sectional Survey Using an Interdisciplinary Healthcare Intervention

**DOI:** 10.3390/healthcare8040382

**Published:** 2020-10-02

**Authors:** Constanta Urzeală, Aura Bota, Silvia Teodorescu, Mihaela Vlăiculescu, Julien S Baker

**Affiliations:** 1Sports and Motor Performance Department, Faculty of Physical Education and Sports, National University of Physical Education and Sports from Bucharest, 060057 Bucharest, Romania; 2Training of Teaching Staff Department, Faculty of Physical Education and Sports, National University of Physical Education and Sports from Bucharest, 060057 Bucharest, Romania; aurabota@ymail.com; 3Doctoral School, National University of Physical Education and Sports from Bucharest, 060057 Bucharest, Romania; teo.silvia@yahoo.com; 4Outpatient Diabetes Clinic “DiabNutriMed” Telemedicine Center, 020358 Bucharest, Romania; vvmihaela@yahoo.com; 5Support for Diabetes Association, 020358 Bucharest, Romania; 6Centre for Health and Exercise Science Research, Department of Sport, Physical Education and Health, Hong Kong Baptist University, Kowloon Tong, Hong Kong, China; jsbaker@hkbu.edu.hk; 7Research Division “Child Public Health”, Department of Child and Adolescent Psychiatry, Psychotherapy and Psychosomatics, Center for Psychosocial Medicine, University Medical Center Hamburg-Eppendorf, 20246 Hamburg, Germany; ravens-sieberer@uke.de

**Keywords:** health-related quality of life, child care, type 1 diabetes

## Abstract

*Background*: The purpose of this study was to assess the quality of life in Romanian type 1 diabetes mellitus (T1DM) children attending an early interdisciplinary healthcare intervention. Hypothesis: engaging T1DM children in leisure sports leads to a better quality of life. *Methods*: This research embeds a cross-sectional observational study, incorporating some clinical characteristics relevant for diabetes management. The Kidscreen 27 questionnaire was issued to 100 T1DM children aged between 7 and 17 years. Parents completed the questionnaire. All subjects received interdisciplinary healthcare in the previous year. Statistics were performed using SPSS, v20. The required sample size of 100 subjects was obtained with a confidence interval of 95% and a sampling error of 0.009. The tests were two-sided, with a type I error set at 0.05. *Results*: Subjects reached an increased level of physical well-being, psychological well-being, autonomy, parent relationships, peer and social support, and school inclusion. There was a significant difference (*p* < 0.05) between children who practice leisure activities and children who only participated in physical education (PE) classes, regarding their physical well-being (t = 2.123). ANOVA demonstrated significant differences between age groups regarding physical well-being. *Conclusion*: The interdisciplinary healthcare intervention increased the efficiency of T1DM management with positive effects on life quality.

## 1. Introduction

The alarming number of people diagnosed with diabetes globally and the negative health predictions related to the increased incidence of this condition make information about the quality of life decisive for this population.

In 2016, the World Health Organization [1] presented a report identifying worrying statistics regarding the incidence of diabetes mellitus (DM) for global populations. The report stated that there were over 400 million people worldwide suffering from this metabolic disorder. The WHO representatives pointed out that there are no separate global estimates outlining the prevalence of type 1 diabetes mellitus (T1DM) and type 2 diabetes mellitus (T2DM).

Children mainly suffer from T1DM, also termed childhood diabetes or juvenile diabetes. This has been identified as a chronic condition without a clear etiology, which involves lifelong dependence on insulin administration. However, the cases of type 2 diabetes mellitus that occur in childhood are becoming more and more numerous, and the research data demonstrate an epidemic increase associated with increases in obesity. In 2019, the International Diabetes Federation [2] recorded over 1.1 million children and adolescents with T1DM worldwide. Specialists estimate that the incidence of this chronic disease has an ascendant trend in children, differing among countries [3]. According to data provided by National Insurance House, about 3500 children were officially diagnosed with this condition in Romania [4].

Different authors emphasize that this autoimmune condition is a permanent challenge for children, adolescents, and their parents, affecting both quality of life and lifestyle of the entire family [5]. Defined as a corollary of physical and psycho-emotional well-being in terms of health and daily functioning, quality of life depends on an individual’s perception of expectations, goals, and concerns experienced within their sociocultural context [6]. As a crucial health outcome index, the quality of life reflects an individual’s self-perception of their personal physical, mental, emotional, social, and behavioral condition [7,8].

Regarding the life quality of children with T1DM, the most important factor is diabetes management, namely, a combination of insulin therapy, diet, and physical exercise, managed during their lifespan and related to daily living activities. An important issue in managing juvenile diabetes is related to the physical pain and inconvenience experienced in each measurement of blood glucose and each insulin administration. This adds to the negative experiences associated with medical procedures, some daily living activities, or diet restrictions. Diabetes management is the responsibility of parents in early childhood; afterwards this is gradually transferred once the child enters adolescence and older ages [9]. Besides specific medical treatment issues, the potential barriers to achieving a good quality of life for children with T1DM are linked to leisure, school involvement, relationships with peers, and the entire psycho-emotional climate of the family environment [10,11]. Along with the congruent efforts of the main healthcare providers (diabetologist, pediatrician, parent, educator, psychologist, social worker) who attend to a child with diabetes, there are also other factors that influence their quality of life, such as: socio-demographic, related to medical history, or specific to different periods of growth and development and psycho-emotional background [12,13].

Physical activity, a constant component within the treatment of diabetes mellitus [14], is in fact a changing variable in the life of a child with this condition. The participation of a child with diabetes in physical exercise programs organized in the form of physical education (PE) classes or leisure activities is largely influenced by blood glucose levels, especially hypoglycemia (defined as a value below 70 mg/dL of blood glucose) [15]. In addition, if the effect of physical exercise on blood glucose is prolonged, the tendency for hypoglycemia lasting for up to 10–12 h after the end of exercise is increased. However, this will depend on both the intensity and duration of exercise and individual metabolic responses to the activity. Fear of hypoglycemic episodes, which are a major immediate risk to a child’s life [16], may prevent them from engaging in physical activities and may explain parental tendency to keep the child’s blood glucose high. At the same time, the risks of hyperglycemia, associated with long-term complications, are less prioritized by the adult, overshadowed by the burden of hypoglycemia [17,18]. Consequently, hyperglycemia will be less often considered in the case of children’s involvement in age-specific activities, compared to hypoglycemia. Because physical exercise is directly associated with hypoglycemia, not all families will support the participation of their children in sports activities, even if, in practice, specific measures are taken to counteract this metabolic reaction [19]. These issues of diabetes management can be controlled by permanent blood glucose monitoring with the help of telemedicine [20], which was gradually introduced in Romania in the private healthcare system in 2015 and in the public system commencing in 2019. Via continuous glucose monitoring (CGM) systems, the family or the older child is constantly informed about blood glucose levels and is also alerted if there is a tendency for hypoglycemia, which makes it possible to intervene before this event occurs.

The child’s need to socialize and make friends is initially achieved through family efforts and interactions via educational networks, including schools and colleges. Moreover, studies have demonstrated that, for young patients with type 1 diabetes mellitus, the support of friends is also a beneficial factor in complying with specialized constraints, including diets which maintain appropriate blood glucose levels [21].

The participation of children with diabetes in school activities is challenging for parents, as children need to attend teaching activities outlined in the schedule, including risk-free PE classes [22]. Attending school activities and at the same time applying diabetes-specific treatment concern not only the family but also the staff that interacts with the child [23]. In Romania, it is common for primary schoolchildren with diabetes to be supervised daily by their parents, including in school settings. Parents are ready to intervene whenever children’s blood glucose levels become health-threatening.

Even if the parent is the central pillar of the child’s life, the adult’s internal psycho-emotional resources that are aimed at managing the stressful life events in diabetes mellitus are not inexhaustible. Specialists often recommend parenting interventions [24] in order to trigger new coping mechanisms and good emotional control [25,26], so that the adult’s psycho-emotional state does not affect the child’s quality of life. In addition, the health status of the child with T1DM and the difficulties in controlling blood glucose levels can negatively influence the family climate, generating a snowballing effect on the child’s psycho-emotional climate [27]. Studies show that there is an increased incidence among people with diabetes to develop mental disorders such as depression, anxiety [28], or eating disorders [29]. The child’s psycho-emotional state, which is perceived by the parent through the child’s mood (apathy, sadness, unhappiness), has a strong impact on the adult, who is willing to make great sacrifices to improve the child’s quality of life, seeking to bring much-needed normality in their daily routines.

The purpose of this paper was to assess the quality of life for children with T1DM attending an early interdisciplinary healthcare intervention that included medical education, nutritional and psychological counseling, and physical activities. In order to emphasize the importance of diabetes management for the quality of life, relevant clinical characteristics decisive for the child’s health status were also analyzed. This framework was designed so that the whole family should be involved in this joint effort. Such a complex approach has been addressed only sporadically in Romania, and is an example of good practice in this area.

Because there is a tendency to remove physical exercise from the individualized intervention plan in order to prevent hypoglycemia, the following hypothesis was established: attending an early interdisciplinary intervention that includes engaging T1DM children in leisure physical activities leads to a better quality of life, compared to those who participate solely in PE classes.

## 2. Materials and Methods

### 2.1. Sample

The study included 100 children with T1DM, treated with multiple daily injections (MDI) or insulin pump therapy, aged between 7 and 17 years old, from Romanian urban areas. All subjects benefited from blood glucose sensors and were constantly monitored by their medical doctor (M.V.) from an outpatient diabetes clinic where the standard of care for children with type 1 diabetes mellitus was based on telemedicine devices.

All subjects received interdisciplinary healthcare for 12 months that consisted of: medical education, nutritional and psychological counseling, and physical exercises. This interdisciplinary healthcare intervention was addressed to both the child and family. The medical care consisted of medical check-ups every three months, rigorous blood glucose control, insulin administration compliance, use of telemedicine tools for monitoring, and glycated hemoglobin tests every three months. Nonetheless, no biological data related to HbA1c testing was collected during this protocol due to the profile of the university coordinating this research and the directions of investigation mentioned within the Research Ethics Commission approval.

Also, the diabetologist organized two conferences for parents and children during the World Diabetes Day, for presenting them with the newest scientific approaches to the management of juvenile diabetes and updating their knowledge. The physical exercises were performed during the school program, being included in the PE classes and in different leisure activities. The school sessions lasted 50 min, two sessions per week, consisting of combined aerobic–anaerobic physical efforts, always monitored by use of glucose sensors and correlated with the blood glucose values. Some of the children also performed leisure activities besides the PE classes, 2–3 sessions per week (swimming, basketball, or dancing) lasting 90 min each. The nutritional and psychological counseling sessions were scheduled one per month, for the entire family. Furthermore, children, parents, and peers participated in two educational camps (one week each) addressing diabetes education overall and social interactions.

For each child involved in this study, one of the parents, aged between 28 and 45 years, was asked to complete an assessment questionnaire about health-related quality of life (HRQOL).

The inclusion criteria for both children and parents included the following: children without severe glycemic excursions and with no need to be hospitalized in the previous year. Thus, the only exclusion criteria were the medical records.

According to the questionnaire application procedure, parents with children under the age of 7 were not included in the survey. Also, 10 from the initial group of parents did not complete the Kidscreen 27 (Figure 1).

Parents’ informed written consent was obtained in accordance with the standards corresponding to research on human subjects and confidentiality of personal data. The study was conducted in accordance with the Declaration of Helsinki, and the protocol was approved by the Research Ethics Commission of the National University of Physical Education and Sports from Bucharest (1582/29.05.2019). This research protocol did not include results of biological samples collected during the 12 months of intervention.

### 2.2. Procedure

From a medical standpoint, the main clinical characteristics intrinsically linked to T1DM and children’ health status were taken into consideration: duration of T1DM, method of insulin administration, percentage of time in hypoglycemia (Hypo) and percentage of time in range—within optimal glycemia values (TIR), per 24 h, at the beginning and at the end of the interdisciplinary intervention (12 months). No biological samples were collected specifically for this protocol. This standard of care provided an objective image of how the children’s health could improve, due to rigorous medical surveillance, including telemedicine monitoring. The glycemia oscillations of the investigated group were continuously monitored by means of sensor-based technology, recognized as a successful tool in routine medical practice. Therefore, an internationally recognized matrix was used in order to properly manage diabetes. The reference taken into consideration for the TIR was 70% from the readings and time spent in 70–180 mL/dl glycemia values [30]. The recorded glycemia values were extracted from the telemedicine reports.

With respect to the quality of life, this research used a cross-sectional observational study design based on identifying the parents’ perspective as the result of the applied interdisciplinary healthcare intervention. In order to achieve the research purpose, the Kidscreen 27 questionnaire was applied as an assessment tool, comprising five areas of investigation, namely, physical well-being (5 items), psychological well-being (7 items), autonomy and parent relationships (7 items), peers and social support (4 items), and school environment (4 items). Kidscreen is a standardized instrument designed to measure the well-being of children and adolescents with or without health problems, which can be completed by children but also by their parents and friends [31]. The Kidscreen 27 questionnaire can be accessed at: https://www.kidscreen.org/english/questionnaires/kidscreen-27-short-version/.

The short version of the Kidscreen 52 questionnaire was used for efficiency reasons. Thus, the Kidscreen 27, which requires only 10 to 15 min completing time, was used in this study. In fact, only Kidscreen 27 was available in Romanian from all the versions validated by the Kidscreen team (versions 52 and 10 items).

Parents were asked to respond in writing by completing a Likert scale from 1 to 5 for each item, within a section specifically dedicated to the interdisciplinary care of children with diabetes. The questionnaires were distributed and collected by volunteers specially trained in this regard, during the World Diabetes Day event. For each area of investigation, responses were converted into points; it should be mentioned that four items required inverting the scoring scale, in accordance with the Kidscreen 27 handbook. Moreover, in order to test the research hypothesis, parents were asked whether their children were medically exempt from physical exercise and whether they participated in sports activities for leisure.

### 2.3. Statistics

Statistical analysis on the data collected was performed using SPSS software, v20 (IBM, Armonk, NY, USA). The required sample size of 100 was obtained with a confidence interval of 95% and a sampling error of 0.009. The tests were two-sided, with a type I error set at *p* < 0.05 (sig.). The points accumulated for each scale of the questionnaire were related to the maximum possible and converted into percentages in order to assess the HRQOL index of the T1DM children. Correlative results were analyzed by means of Pearson’s linear correlation coefficient (r) and Spearman correlation between paired data. The *t*-test (t) was applied to identify the statistical significance between certain clinical characteristics, physically active (PA) and physically inactive (PI) children, as well as the differences between age groups and testing moments (at the beginning and after the intervention). Using Levene’s test, it was possible to verify if the groups had equal or different variances [32]. Analysis of variance (ANOVA) was performed to consider the children’s age differences and participation in leisure sports activities. Thus, the subjects were divided into four groups: 26 PA under 10 years old, 24 PI under 10 years old, 24 PA over 11 years old, 26 PI over 11 years old. Given the sizes of these groups, Gabriel’s post hoc test was used to identify if one variable differed from the other [33].

## 3. Results

### 3.1. Analysis of Clinical Characteristics

Table 1 emphasizes the relevant measured outcomes, statistically processed in terms of central tendency. Duration of T1DM, method of insulin administration, time in hypoglycemia, and time in range were the outcome measures analyzed in their dynamics across one year of interdisciplinary intervention. In addition, also included are the age, gender, and the active/non-active behavior of the T1DM children.

According to these data, the group comprised an equal number of children aged between 7–10 years old (50) and children aged between 11–17 years old (50), and a perfectly balanced proportion between girls and boys. From the whole group, 50% were physically nonactive, while 50% were physically active.

Regarding the clinical characteristics, the central tendency measures showed a mean value of 2.92 years for the duration of T1DM, a percentage of 60% of MDI for insulin administration, a percentage of 40% of insulin-pump-treated patients. CGM data showed a percentage of 5.53% out of 24 h spent in hypoglycemia, at the beginning of the interdisciplinary intervention, and 3.73% out of 24 h spent in hypoglycemia, after one year of the intervention. The mean percentage of TIR at baseline was 64.33%, while the same outcome measured after 12 months of intervention registered an increase by 6.62%.

Using *t*-test for paired samples, a significant difference was identified between time spent in hypoglycemia prior to and after the intervention t(99) = 7.193, *p* = 0.000, with a large size effect (d = 0.89); df represents the degrees of freedom. There was also a significant difference for TIR outcome between prior and post intervention t(99) = 16.20, *p* = 0.000, with a large effect size (d = 0.78) (Table 2).

Regarding the TIR at baseline, the *t*-test applied for independent samples (insulin pump administration versus MDI) demonstrated a significant difference (*p* < 0.001) between the children receiving MDI and those receiving insulin pump therapy [t(98) = 7.273]. The effect size was large (d = 1.48). After the interdisciplinary intervention was administered, the *t*-test for TIR revealed a significant difference (*p* < 0.001) between children receiving MDI and those who were provided with insulin pump treatment [t(98) = 7.763]. The effect size was large (d = 1.58) (Table 3).

With respect to the percentage of time spent in Hypo at baseline, the *t*-test applied for independent samples (insulin pump administration versus MDI) demonstrated no significant difference (*p* > 0.001) between the children receiving MDI and those receiving insulin pump therapy. The same statistical test applied for analyzing the percentage of time spent in Hypo at 12 months revealed no significant difference (*p* > 0.001) between the MDI group and the insulin pump group.

### 3.2. Analysis of Well-Being Parameters

Table 4 shows the descriptive statistics of the questionnaire scales, which illustrated that all analyzed variables were normally distributed. The absolute skewness coefficient was less than 1 [34].

Figure 2 highlights that subjects with T1DM participating in this study reached an increased level for all parameters (over 71%), which indicated a good quality of life for these children. The lowest values (71.45%) were recorded for the social support provided by friends, namely, a percentage of 71.45%.

Pearson’s linear correlation coefficient revealed medium and high correlations (*p* < 0.01) between the different components that ensure quality of life for the investigated subjects (Table 5).

### 3.3. Physically Active Children with T1DM versus Physically Inactive Children with T1DM

Children were divided into two groups according to their participation in leisure activities. One group included physically active (PA) children: 50 subjects (26 boys and 24 girls) aged between 7 and 17 years, with a mean age = 10.79 and standard deviation = 2.80. The other group included physically inactive (PI) children: 50 subjects (24 boys and 26 girls) aged between 7 and 17 years, with a mean age = 10.52 and standard deviation = 2.75. No subjects were exempt from physical effort for medical reasons, which meant that all subjects participated in PE classes.

To verify whether there were statistically significant differences in quality of life between the two groups, *t*-tests for independent samples were applied (Table 6).

According to the *t*-test results, there was a significant difference (*p* < 0.05) between children who practiced leisure activities and children who did not practice sports in their free time. Only physical well-being components were assessed by parents (t = 2.123). However, there was no effect size for this aspect (d = 0.001) [35]. The physical well-being component had higher values in the group of children who practiced sports in their free time (M = 18.76, SD = 3.55), compared to children who participated only in PE classes (M = 17.22, SD = 3.69). For psychological well-being, autonomy and parent relationship, peers and social support, school environment and overall quality of life, there were no statistically significant differences (*p* > 0.05).

As regards the age variable, its correlation with the questionnaire scales was analyzed (Table 7), and there was a single significant correlation (*p* < 0.05) found between age and the physical well-being scale. However, this correlation was minor [36].

Regarding the age and participation in leisure sports activities, the ANOVA analysis table (Table 8) displays, for each source of variance, the following: sum of squares, degrees of freedom (df), mean squares, test scores, *p* significance thresholds (Sig.), effect size (Eta partial square). The table’s note refers to the R^2^ and adjusted R^2^ coefficients for the explanatory model based on the regression method.

ANOVA demonstrated that there were significant differences between the four groups (PA under 10 years old, PI under 10 years old, PA over 11 years old, PI over 11 years old) in terms of physical well-being, F (3.96) = 3.822, at *p* < 0.05. The effect size (η^2^p = 0.107) was average [35]. Levene’s test recorded an insignificant value (*p* = 0.318), which showed that the dispersions within the four groups were homogeneous.

Gabriel’s post hoc test showed that there were differences between groups, but the only significant difference was between the PA group including children under 10 years old and the PI group comprising children over 11 years old (Table 9).

Statistical analysis showed that physical well-being had the highest level in PA children under 10 years old (M = 19.65, SD = 3.32). The lowest values were recorded for PI children with T1DM over 11 years old (M = 16.35, SD = 4.04).

After analyzing data regarding the quality of life, statistical correlations were performed taking into account the clinical characteristics of study participants.

Among all correlations between clinical characteristics, Kidscreen total score, and physically active/nonactive behavior, there were identified five significant relationships observed at *p* = 0.000: TIR at baseline and method of insulin administration, TIR at 12 months and method of insulin administration, TIR at 12 months and duration of T1DM, Hypo at 12 months and duration of T1DM, TIR at 12 months and Kidscreen total score (Table 10).

## 4. Discussion

Because of improvements in technology accuracy, medical commitment to novel approaches, and national legislation enforcing expenditure reimbursement of the related costs, a greater access for Romanian T1DM children to the CGM systems became possible as an important prerequisite for an improved clinical care and higher quality of life. This study emphasized interesting results concerning both diabetes management outcomes and life quality dimensions, as perceived by the parents.

In terms of time in hypoglycemia per 24 h, the interdisciplinary intervention led to a better prevention of under 70 mg/dL glucose values. This might be explained by a balanced insulin administration, a proper nutrient distribution across 24 h, and an adequate selection of physical activities performed by the subjects. Furthermore, the therapeutic education on prevention and treatment of exercise-related hypoglycemia surely represented an essential tool for obtaining these results.

Significant differences were also found between the prior and post interventions in terms of TIR proving that the combined effects of the medical standard of care and the other educational means used in the interdisciplinary approach diminished the time spent in hypoglycemia, improving long-term prognosis of the patient. With respect to the time in range, the children receiving multiple daily injections had a lower time in range compared to those receiving insulin pump therapy, prior to and after the intervention. Focusing on the statistically significant correlations among the outcome measures, this study demonstrated that the duration of T1DM correlated with the post intervention clinical characteristics, namely, increased time spent in hypoglycemia as going further from the diagnosis moment. There was also a moderate positive correlation between the duration of T1DM and TIR at 12 months. Although it is well known that a perfect metabolic balance is difficult to achieve as going further from the diagnosis moment, the current study showed that, for the investigated children, this health-related status became possible to attain in a complex diabetes management intervention based on medical care, a regular educational program, and the use of CGM.

Other significant correlations were identified between TIR at baseline and TIR at 12 months, and the method of insulin delivery, confirming the advantages of CGM devices accompanied by insulin pump therapy compared with MDI. Either way, the glucose excursions are likely to be reduced due to the real-time glucose value visualization and the possibility of early detection of hypoglycemia [37]. Nonetheless, the insulin delivery method did not influence time spent in hypoglycemia, the group with MDI registering similar results with the insulin pump therapy group at the end of the intervention. The Kidscreen total score significantly correlated with the time in range post intervention, regardless of insulin delivery method. Therefore, we may assume that an early detection of T1DM combined with a highly proficient standard of care are determinant factors for proper management of this metabolic disease, by strengthening the quality of life dimensions for all children experiencing this medical condition.

The development of Kidscreen instruments in 13 European countries [31] has stimulated interest in its use in Romania as well, in order to assess quality of life for children with chronic diseases. In this context, the current study is the first to demonstrate the effectiveness of using the Kidscreen 27 questionnaire to investigate children with diabetes in Romania. This aspect must be highlighted along with the replicability and completion of this approach with other survey instruments [38], to minutely analyze the life quality in children with diabetes compared to healthy children [12].

In this context, the present study identified a good quality of life for Romanian children with T1DM who benefited from an early interdisciplinary healthcare intervention. The need to monitor the quality of life for children with diabetes has been demonstrated by authors such as Morillo et al. [39]. Being associated with the management of this metabolic disorder, we can state that patients with diabetes participating in this research benefited from appropriate healthcare. Similar studies were conducted by Hoey et al. [40] on adolescents from Japan, North America, and European countries.

The study highlighted that there was a group of children with T1DM in Romania who enjoyed an increased level of physical well-being, psychological well-being, autonomy and parent relation, peers and social support, and school environment inclusion. Outcomes’ progress was due to the applied interdisciplinary healthcare intervention based on the education of the family regarding not only insulin treatment and diet, but also the role of physical exercise in maintaining the physical and mental health of the child with this chronic condition. Generally speaking, medical education is the focus of this healthcare intervention where the diabetologist prescribes the treatment, recommends nutrition and physical effort in accordance with blood glucose values, and has the greatest influence on the family and the patient lifestyle. In addition, the psychological counseling of parents and children regarding the fear of hypoglycemia made these families open-minded toward the practice of sports activities. The physical exercises programs provided by specialists led to engaging children in both PE classes and leisure activities. These children and their families can be examples of good practice in the management of childhood diabetes, knowing that the need for peer support and the need for additional information about the life of a child with diabetes are scientifically proven [41]. This research is also an evidence-based approach to socially supporting the children with chronic diseases and their families, who often face labeling, marginalization, and even exclusion. This endeavor requires a joint effort of different social sectors like government entities, medical institutions, schools, communities, educational organizations, and sports clubs.

### 4.1. Physical Well-Being

The research results revealed that the participating subjects had an optimal perceived level of fitness, showing energy in their free time for either play, domestic activities, or sports, without any complaints about their health status.

This approach scientifically validated once again the benefits of physical activity for the child with diabetes and contributed to reducing the parents’ reluctance and opposition to letting their children engage in physical exercise programs despite the risk of hypoglycemia. The research hypothesis was only partially confirmed, the results showing that, in the case of the investigated group of subjects, children with T1DM who practiced sports activities in their free time had an increased level of fitness, in the view of parents, compared to those who were engaged only in PE classes. Psychological well-being, autonomy and parent relation, peers and social support, school environment and HRQOL index did not differ between children with diabetes who practiced sports in their free time and those who only attended school physical education.

The adults’ constant struggle to maintain good blood glucose levels for their children and fear of hypoglycemia [42] influenced participation in sports activities. However, the combined advantages of practicing leisure sports, from metabolic regulation to psycho-emotional balance, were reflected in diabetes management and the child’s quality of life [43]. In this context, the research results support the favorable influence of physical activity, demonstrating that there is a positive correlation between physical well-being and psychological well-being in children with T1DM.

It has also been shown that at young ages, during primary school, children with diabetes who practice sports in their free time have a better physical condition than physically inactive children in middle school and high school. This outcome is achieved in the context wherein the preference for a sedentary lifestyle, which occurs at puberty and in adolescence, is prevalent also in healthy children [44].

### 4.2. Psychological Well-Being

Results for the psychological well-being component indicated that the participating subjects were perceived by their parents as expressing positive emotions and being satisfied with their lives, without feeling sad or lonely. Moreover, this research demonstrated that the psychological well-being of children with T1DM correlates with all the components of quality of life. Although studies showed that patients with T1DM are more prone to mental disorders compared to other categories of population [45], at the time of applying the research questionnaire, parents believed that their children managed to cope with the burden of diabetes without perceiving psycho-emotional disorders. However, it is worth noting that only a specialist is able to assess the need for a psychotherapeutic intervention.

### 4.3. Autonomy and Parent Relation

Results for the autonomy and parent relation component highlighted that children interacted well with their parents and enjoyed their support and love. It can be stated that the participating children with T1DM belonged to harmonious families, able to manage diabetes, diet, and daily living activities. Other studies emphasized that conflicting family relationships, lack of an organized lifestyle, and parents’ emotional imbalance negatively influence the children’s overall status [46] and thus their quality of life.

### 4.4. Peers and Social Support

Belonging to a group of friends with whom children can share opinions, leisure activities, experiences, preferences, and diabetes-related concerns, in accordance with their age and gender, is helpful to integrate diabetes into one’s own self-identity [47]. This idea is also reflected in the higher scores recorded by all subjects participating in this research for social support as a component of quality of life. Surveyed parents believed that their children interacted well with peers and enjoyed support from friends and classmates. The analysis highlighted correlations between peers and social support, autonomy and parent relation, and school environment components. It should be noted that the group of subjects belonged to a community that included young adults with T1DM who practiced performance sports and were regarded as role models for children and parents. Therefore, community support is extremely important in diabetes management.

### 4.5. School Environment

It has been demonstrated that there is a need to support school in different countries [48] as a complementary trusted educational setting. The authors of this paper, who are involved in working with both children suffering from T1DM and their families, consider that Romania is facing the same problem in relation to an in-depth knowledge of childhood diabetes. However, adult respondents perceived child participation in school learning activities as positive. They believed that children had good feelings about school and the ability to focus and learn in the context of good understanding and positive communication with their teachers.

Lack of studies investigating quality of life in children with diabetes through the Kidscreen questionnaire in Romania hindered the extrapolation and generalization of the results.

The number of subjects was influenced by the moderate incidence of childhood diabetes in Romania and by the complicated bureaucratic procedures to get access to this category of population. It was also the result of parents’ reluctance to become involved in scientific research, which inevitably led to low participation in this study.

The fact that this research included only children who benefited from real-time blood glucose monitoring was related to the profile of the medical center they attended, namely a healthcare clinic supporting telemedicine for children with diabetes.

Lack of a longitudinal design for the surveyed data was another limitation of this study. The study was cross-sectional in nature and focused on revealing parents’ opinions as a reference for specialists who will further improve interdisciplinary interventions. This snapshot study was also chosen because parents of T1DM children usually complete massive documentation related to constant monitoring, which is time-consuming and overwhelming, their primary concern being to stabilize glycemia oscillations. Under these circumstances the parents were asked to complete the questionnaire only after the interdisciplinary intervention was implemented.

## 5. Conclusions

Early interdisciplinary healthcare increases the efficiency of T1DM management and therefore the children’s quality of life. Medical education of both family and child must begin with the onset of the disease and be complemented by nutritional and psychological counseling, along with physical activities, as the main components of a complex healthcare intervention requiring an interdisciplinary team.

Supporting an objective argument on the quality of life of T1DM children, the clinical characteristics provided an estimate of several health-related outcomes that are pivotal in maintaining a risk-free daily existence for this category of population.

Parents’ perspectives on the life quality of their children with diabetes stand as a feedback for all those involved in childcare, from both a medical and a psycho-emotional position. Assessing the T1DM child’s quality of life generates not only mechanisms of family resilience and behavioral regulation, but also certain benchmarks to adjust treatment.

It should be emphasized the importance of engaging children with diabetes in leisure sports activities as part of a complex healthcare strategy so that they fully enjoy physical and psychological well-being as key factors for improving their life quality.

The participation of children with T1DM in school activities is closely related to both the support of family and friends, and the general atmosphere created by classmates, teachers, nurses, and other members of the school staff who become active actors in ensuring the quality of life for these pupils.

Social support from friends and classmates, as well as family inclusion into a community advocating for children with special needs, are reflected in positive communication, sensitivity, and responsibility at macro-social level. Thus, enhancing the life quality and integration of these children into wider communities becomes the ultimate outcome of an advanced society.

## Figures and Tables

**Figure 1 healthcare-08-00382-f001:**
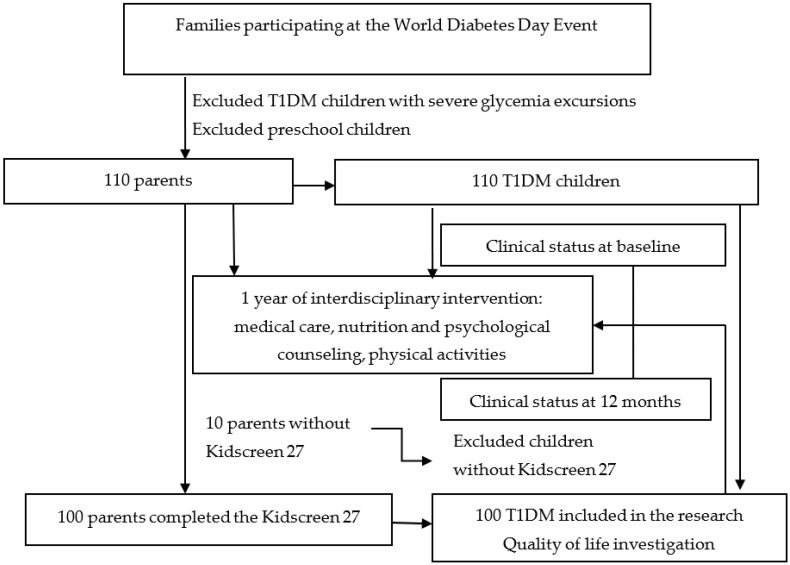
Research protocol flowchart.

**Figure 2 healthcare-08-00382-f002:**
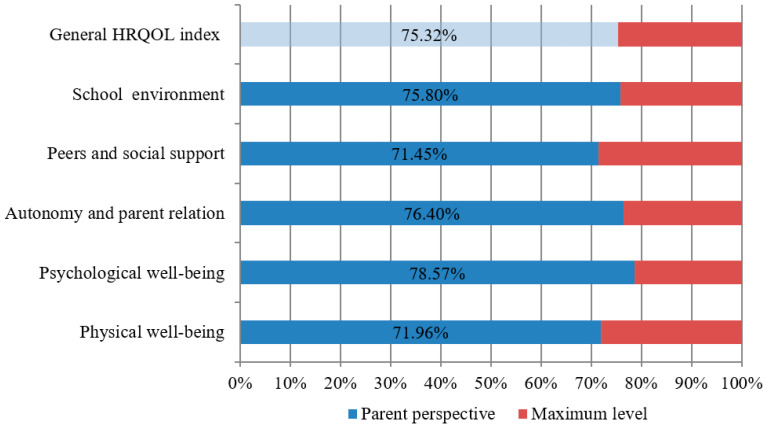
Parent perspective on the quality of life of children with T1DM participating in the study.

**Table 1 healthcare-08-00382-t001:** Descriptive statistics of the measured outcomes.

Age and Active Behaviour	*N*	Percent	Mean
Aged between 7–10 years old	50	50.0%	
Aged between 11–17 years old	50	50.0%	
Girls	50	50.0%	
Boys	50	50.0%	
Leisure:			
Non-active children	50	50.0%	
Active children	50	50.0%	
Clinical characteristics			
Duration of type 1 diabetes mellitus (T1DM)	100		2.92 years
Method of insulin administration			
Insulin pump	40	40.0%	
Multiple daily injections (MDI)	60	60.0%	
%Time in hypoglycemia/24 h at baseline			5.53%
%Time in hypoglycemia/24 h at 12 months			3.73%
%Optimal glycemia values (TIR)/24 h at baseline (whole group)			64.33%
%TIR/24 h at baseline insulin pump group			71.50%
%TIR/24 h at baseline MDI group			59.55%
%TIR/24 h at 12 months (whole group)			70.95%
%TIR/24 h at 12 months insulin pump group			76.07%
%TIR/24 h at 12 months MDI group			67.53%

**Table 2 healthcare-08-00382-t002:** *T*-test for %time spent in hypoglycemia and time in range per 24 h.

	Paired Differences					t	df	Sig. (2-Tailed)
	Mean	Std. Deviation	Std. Error Mean	95% Confidence Interval of the Difference				
				Lower	Upper			
Hypo at the beginning Hypo at the end	1.800	2.503	0.250	1.303	2.297	7.193	99	0.000
TIR at the beginning TIR at the end	−6.620	4.084	0.408	−7.430	−5.810	−16.20	99	0.000

**Table 3 healthcare-08-00382-t003:** *T*-test for TIR and Hypo–insulin pump versus MDI.

	Levene’s Test		*T*-Test				
	F	*p*	t	df	*p*	Confidence Interval	
						Lower	Upper
TIR at baseline	0.415	0.521	−7.273	98	0.000	−11.950	1.643
TIR at 12 months	0.066	0.797	−7.763	98	0.000	−8.542	1.100
Hypo at baseline	5.643	0.019	−1.465	96.99	0.146	−0.742	0.506
Hypo at 12 months	0.102	0.750	−1.482	98	0.142	−0.325	0.219

**Table 4 healthcare-08-00382-t004:** Descriptive statistics of the questionnaire scales for the whole sample.

	*N*	Min	Max	M	SD	Symmetry Statistic Standard Error		Flattening Statistic Standard Error	
Physical well-being	100	11	25	17.99	3.69	−0.071	0.241	−0.792	0.478
Psychological well-being	100	16	35	27.50	4.01	−0.203	0.241	−0.390	0.478
Autonomy and parent relation	100	12	35	26.74	5.04	−0.533	0.241	−0.326	0.478
Peers and social support	100	8	20	14.29	3.14	−0.048	0.241	−0.708	0.478
School environment	100	8	20	15.16	2.76	−0.272	0.241	−0.385	0.478
Health-related quality of life (HRQOL) index	100	65	133	101.68	14.81	−0.425	0.241	−0.198	0.478

Note: number of subjects (*N*), maximum values (Max), minimum values (Min), means (M), standard deviations [1], symmetry indicators (standard error skewness) and flattening indicators (standard error kurtosis).

**Table 5 healthcare-08-00382-t005:** Pearson’s correlations between the components of quality of life.

		Psychological Well-Being	Autonomy and Parent Relation	Peers and Social Support	School Environment
Physical well-being		0.659 **	0.479 **	0.376 **	0.449 **
	Sig.	0.000	0.000	0.000	0.000
	*N*	100	100	100	100
Psychological well-being			0.600 **	0.503 **	0.542 **
	Sig.		0.000	0.000	0.000
	*N*		100	100	100
Autonomy and parent relation				0.595 **	0.539 **
	Sig.			0.000	0.000
	*N*			100	100
Peers and social support					0.516 **
	Sig.				0.000
	*N*				100

** Correlation is significant at the 0.01 level.

**Table 6 healthcare-08-00382-t006:** *T*-test for independent samples.

	Levene’s Test		*T*-Test				
	F	*p*	t	df	*p*	Confidence Interval	
						Lower	Upper
Physical well-being	0.741	0.392	2.123	98	0.036	1.54	0.72
Psychological well-being	1.547	0.217	1.299	98	0.197	1.04	0.80
Autonomy and parent relation	0.520	0.472	−0.632	98	0.529	−0.64	1.01
Peers and social support	0.256	0.614	1.180	98	0.241	0.74	0.62
School environment	0.088	0.767	−0.577	98	0.565	−0.32	0.55
HRQOL	0.541	0.464	0.795	98	0.429	2.36	2.96

**Table 7 healthcare-08-00382-t007:** Correlations between age and questionnaire scales.

		Physical Well-Being	Psychological Well-Being	Autonomy and Parent Relation	Peers and Social Support	School Environment	HRQOL
Age		−0.251 *	−0.189	−0.130	−0.030	−0.097	−0.183
	Sig.	0.012	0.059	0.197	0.769	0.336	0.069
	*N*	100	100	100	100	100	100

* Correlation is significant at the 0.05 level.

**Table 8 healthcare-08-00382-t008:** ANOVA (analysis of variance) in the scores for physical well-being according to the age and participation in leisure sports activities.

Source	Type III Sum of Squares	df	Mean Square	F	Sig.	η^2^p
Corrected Model	143.929 ^a^	3	47.976	3.822	0.012	0.107
Intercept	32,310.731	1	32,310.731	2574.003	0.000	0.964
Group4	143.929	3	47.976	3.822	0.012	0.107
Error	1205.061	96	12.553			
Total	33,713.000	100				
Corrected Total	1348.990	99				

Note: ^a^ R-squared = 0.107 (Adjusted R-squared = 0.079).

**Table 9 healthcare-08-00382-t009:** Results for Gabriel’s post hoc test.

	Physically Inactive (PI) under 10	Physically Active (PA) over 11	PI over 11
PA under 10	1.49	1.86	3.31 *
PI under 10		0.37	1.82
PA over 11			1.45

* Correlation is significant at the 0.05 level.

**Table 10 healthcare-08-00382-t010:** Correlations among clinical characteristics and Kidscreen total score.

		Spearman Correlations			Pearson Correlations		
		Kidscreen Score	Leisure	Method of Insulin Administration	Kidscreen Score	Hypo at 12 Months	TIR at 12 Months
Duration of T1DM			0.150		0.134	0.360 **	0.421 **
	Sig.		0.135		0.184	0.000	0.000
	*N*		100		100	100	100
Method of insulin administration		0.041	0.122				
	Sig.	0.683	0.225				
	*N*	100	100				
Hypo at baseline				0.171			
	Sig.			0.089			
	*N*			100			
Hypo at 12 months			0.093	0.148	0.064		
	Sig.		0.356	0.142	0.526		
	*N*		100	100	100		
TIR at baseline				0.587 **			
	Sig.			0.000			
	*N*			100			
TIR at 12 months			0.74	0.622 **	0.258 **		
	Sig.		0.465	0.000	0.000		
	*N*		100	100	100		

** Correlation is significant at the 0.01 level.

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
