# Peer review of "Quality of Life in Romanian Children with Type 1 Diabetes: A Cross-Sectional Survey Using an Interdisciplinary Healthcare Intervention"

_healthcare, 2020, doi:10.3390/healthcare8040382_

Round 1
Reviewer 1 Report
The missing informations have been added.
Reviewer 2 Report
I want to express my gratitude for the opportunity to re-review the manuscript entitled "Quality of Life in Romanian Children with Type 1 Diabetes: A Cross-Sectional Survey in an Interdisciplinary Healthcare Intervention" (healthcare-955943, resubmission of healthcare-909765) by Constanta Urzeală et al. The authors revised the manuscript profoundly and took into account my previous comments regarding the study design, methodology, presentation of results and the discussion. Therefore, in my opinion, the manuscript meets the criteria necessary for publication in Healthcare.
This manuscript is a resubmission of an earlier submission. The following is a list of the peer review reports and author responses from that submission.
Round 1
Reviewer 1 Report
The aim of the study was to assess the quality pf life of T1D children, from 7 to 17 years old.
It would have been great :
- to provide a link to an example of the kidscreen 27 questionnaire, especially for non specialists. So that the criterion for analysis be clearer.
- always for non specialists, provide the p value, when possible
Overall, despite a quite small number of patients (100), this work provides interesting results that may support further studies.
Reviewer 2 Report
Reviewer’s report
Title: Quality of Life in Romanian Children with Type 1 Diabetes: A Cross-Sectional Survey in an Interdisciplinary Healthcare Intervention
Authors: Constanta Urzeală, Aura Bota, Silvia Teodorescu, Mihaela Vlăiculescu, Julien S Baker, The Kidscreen Group Europe
General comment:
In their work, Urzeală et al. used the Kidscreen 27 questionnaire to assess the quality of life in 100 Romanian T1DM children following an early interdisciplinary healthcare intervention. They found that this kind of intervention increased the efficiency of T1DM management with positive effects on life quality, especially in physically active children. The study hypothesis and the conclusions drawn are important for clinical practice, and the manuscript is well written. However, before it is accepted for publication in the Healthcare journal, some significant issues should be considered.
Major revisions:
Study design
The application of the Kidscreen 27 questionnaire “aims to identify children at risk, in terms of their subjective health, and suggest appropriate early interventions by including the instrument in health services research and health reporting” (https://www.kidscreen.org/). Therefore, optimally the study should compare the results of the Kidscreen 27 questionnaire before and after the interdisciplinary healthcare intervention. Even though the Authors mention the lack of longitudinal design as a limitation of the study, this aspect should be wider discussed.
Results
Even though the study focuses on the quality of life of study participants, their clinical characteristics should be presented in, e.g., a separate table including not only age, gender, and information regarding physical activity, but also information on disease duration, age at diagnosis, age at intervention, method of insulin administration (multiple injections vs. insulin pomp), mean HbA1c levels, fasting, and postprandial glycemia, etc. An attempt to correlate clinical data with the results of the Kidscreen 27 questionnaire could also be performed.
Minor revisions:
Abstract
The primary hypothesis regarding the positive influence of physical activity on quality of life should be more clearly presented in the abstract, making the conclusions more understandable.
Materials and methods
2.1 Sample
It is not clear how many children were initially included into the study: in lines 139-140 the authors state “The study included 100 children with diabetes (girls and boys) aged between 7 and 17 years, from Romanian urban areas”; however in line 161: “Also, ten parents did not complete the questionnaire" subsequent statistical analyses involve 100 study participants. These details could be, for instance, presented on a graph with a study protocol. Besides, the inclusion and exclusion criteria for the study could be presented more clearly.
